# Prevalence of post-traumatic stress disorder on health professionals in the era of COVID-19 pandemic, Northwest Ethiopia, 2020: A multi-centered cross-sectional study

**Sintayehu Asnakew**[1]*, **Getasew Legas**[1], **Tewachew Muche Liyeh**[2], **Amsalu Belete**[1], **Kalkidan Haile**[3], **Getachew Yideg Yitbarek**[4], **Wubet Alebachew Bayih**[5], **Dejen Getaneh Feleke**[5], **Binyam Minuye Birhane**[5], **Haile Amha**[6], **Shegaye Shumet**[7], **Ermias Sisay Chanie**[5]

**1** Department of Psychiatry, School of Medicine, College of Health Science, Debre Tabor University, Debre Tabor, Ethiopia, **2** Department of Midwifery, College of Health Science, Debre Tabor University, Debre Tabor, Ethiopia, **3** Department of Psychiatry, Debre Markos Referral Hospital, Debre Markos, Ethiopia, **4** Department of Biomedical Sciences, College of Health Science, Debre Tabor University, Debre Tabor, Ethiopia, **5** Department of Pediatrics and Child Health Nursing, College of Health Science, Debre Tabor University, Debre Tabor, Ethiopia, **6** Department of Psychiatry, College of Health Science, Debre Markos University, Debre Markos, Ethiopia, **7** Department of Psychiatry, College of Medicine and Health Science, University of Gondar, Gondar, Ethiopia

* sintie579@gmail.com

## Abstract

### Objective

This study aimed to assess the prevalence and associated factors of post-traumatic stress disorder among health professionals working in South Gondar Zone hospitals in the era of the COVID-19 pandemic, Amhara Ethiopia 2020.

### Methods

Institutional based cross-sectional study design was conducted. A total of 396 respondents completed the questionnaire and were included in the analysis. A previously adapted self-administered pretested standard questionnaire, Impact of Event Scale-Revised (IES-R-22) was used to measure post-traumatic stress disorder. Data was entered into Epi data version 4.4.2 then exported to SPSS version 24 for analysis. Descriptive and analytical statistical procedures, bivariate, and multivariate binary logistic regressions with odds ratios and 95% confidence interval were employed. The level of significance of association was determined at a p-value < 0.05.

### Results

The prevalence of post-traumatic stress disorder among health care providers in this study was 55.1% (95% CI: 50.3, 59.6). Lack of standardized PPE supply (AOR = 2.5 7,95CI;1.37,4.85), respondents age > 40 years (AOR = 3.95, 95CI; 1.74, 8.98), having medical illness (AOR = 4.65, 95CI;1.65,13.12), perceived stigma (AOR = 1.97, 95CI;1.01, 3.85),

**Data Availability Statement:** All relevant data are within the manuscript and its Supporting information files.

**Funding:** The authors received no specific funding for this work.

**Competing interests:** The authors declared no competing interest.

history of mental illness(AOR = 8.08,95IC;2.18,29.98) and having poor social support (AOR = 4.41,95CI;2.65,7.3) were significantly associated with post-traumatic stress disorder at p-value < 0.05. Conversely, being a physician (AOR = 0.15, 95CI; 0.04, 0.56) was less affected by PTSD.

## Conclusions

The prevalence of post-traumatic stress disorder among health care providers in this study was high. Adequate and standardized PPE supply, giving especial emphasis to those care providers with medical illness, history of mental illness, and having poor social support, creating awareness in the community to avoid the stigma faced by health care providers who treat COVID patients is recommended.

## Introduction

The COVID-19 pandemic is the largest outbreak [1] that initially was seen at the end of December 2019 in China, Wuhan city [2]. Within a short time, the number of cases has radically increased within and beyond China, and WHO declared the COVID-19 outbreak as a pandemic [3]. In Ethiopia, the first COVID-19 cases were reported on 13 March 2020. To prevent the spread of the pandemic the government interrupted schools, restricted public assembly and mass transport, ordered civil servants who had a chronic illness to work from home, and closed borders. The government also banned flights to countries, restricted mass transport and declared a five-month national state of emergency, and officially postponed the election for unspecified periods. All these circumstances make, people feel afraid, worried, anxious, and depressed [4].

Since health care workers are involved in the direct care of patients, they are more likely to be infected than the general population [5]. This makes them fear of contagion, concern for family health, interpersonal isolation, trust in and support from their organization, information about risks, and stigma [6–8]. Consequently, health professionals are under overwhelming psychological pressure, which may lead to various psychological problems, such as post-traumatic stress disorder, fear, depression, and insomnia [9].

As it has been shown by different research, greater numbers of health care workers are at risk of developing posttraumatic stress disorder (PTSD) and posttraumatic stress symptoms (PTSS). In a study done in China during the initial phase of the COVID-19 outbreak, more than half of the respondents were psychologically affected as moderate-to-severe [10]. Different studies done in Italy showed that the prevalence of PTSD was 24.73% [11], 43% [12], 49.38% [13], 39.8% [14]. Similarly, the research done in Taiwan on nurses who worked during the outbreak of SARS showed that 11% of the nurses had stress reaction syndrome [15]. Moreover, the studies conducted on mental health outcomes of the COVID-19 pandemic in the United States and Toronto hospitals revealed that the prevalence of post-traumatic stress disorder was 22.8% [16], and 13.8% [17] respectively. Likewise, a systematic review and meta-analysis studies showed that the estimated pooled prevalence of post-traumatic stress disorder on health workers was found to be 21.5%, 31.4%, and 26.9% [18–20]. Moreover, in studies done in Norway and Greece among health workers during the COVID-19 outbreak, the prevalence of PTSD was 28.9% [21] and 16.7% [22] respectively. The study conducted in Wuhan, China revealed that PTSD symptoms occurred among 31.6% of health care providers [23].

According to previously published research works on health care workers, the risk of developing post-traumatic stress disorder was affected by exposure level, working role, years of work experience, social and work support, quarantine, age, gender, marital status, and coping styles, healthcare worker stigma by the community [11, 13, 16, 24].

Studying the effects of COVID-19 on the mental health of health care professionals is important to provide baseline data for health care managers for early screening of the mental health status of health professionals.

Additionally, conducting research on this area is critically needed to provide scientific evidence for the development of prevention and treatment strategies for mental problems during the present as well as future pandemics.

Therefore, the study aimed to assess the prevalence and associated factors of post-traumatic stress disorder on health professionals in the ear of the COVID-19 pandemic in South Gondar Zone hospitals, Amhara, Ethiopia, 2020.

## Materials and methods

### Study setting and period

A multi-centered institutional-based cross-sectional study was conducted at South Gondar Zone, hospitals, Amhara, from April up to May 2020. South Gondar is about 666 km north of the capital city of Addis Ababa. There are 8 hospitals in this zone which include Debretabor general hospital, Andabet, Estie, Addis Zemen, Ebnat, Lay Gaynt, Tach Gaynt, and Simada primary hospitals. There are about 736 health professionals currently serving these hospitals. Mental health care is one of the services rendered within these hospitals.

### Sample size determination

We determined the sample size by using the single population proportion formula with the assumptions of the prevalence of post-traumatic stress disorder = 50% (as there were no studies done in Ethiopia in this area), 1.96 Z (standard normal distribution), α = 0.05, and 10% non-response rate. Accordingly, a representative sample was calculated to be 423.

### Sampling technique, study participants, and participating hospitals

This study was conducted among health professionals working in eight South Gondar Zone governmental hospitals. There are 736 health professionals in these hospitals. Health professionals were from Debretabor (N = 325), Andabet (N = 62), Estie (N = 55), Addis Zemen (N = 75), Ebnat (N = 45), Lay Gaynt (N = 69), Tach Gaynt (N = 53), Simada (N = 52).

All health professionals working in South Gondar Zone hospitals fulfilled the inclusion criteria, and those participants who were on annual leave and severely ill were excluded. We proportionally allocated the sample size to each hospital and we invited 423 participants by using a simple random sampling technique. Of these, twelve (12) of the eligible participants refused to participate and five (5) of the questionnaires were discarded because of incomplete data. Finally, 396 participants completed the questionnaires and were included in the analysis.

### Study variables

The dependent variable was post-traumatic stress disorder measured by the 22 items of the Impact of Event scale-22 (IES-R-22). We measured post-traumatic stress disorder as a dichotomous variable (yes/ no). Independent variables include socio-demographic factors (age, gender, ethnicity, marital status, religion, profession, and having children), clinical variables (family history of medical illness, history of mental illness, having medical illness, psychosocial

and material factors consist of social support, perceived stigma and lack of adequate and standardized PPE supply.

## Data sources, measurement and operational definitions

Data was collected by standardized self-administered questionnaires by 16 trained health professionals using the Amharic version of the tool. The questionnaire was designed in English and translated to Amharic and back to English to maintain its consistency. Data collectors were trained on how to collect the data and explain the unclear questions and the purpose of the study. Furthermore, they were made aware of ethical principles, such as confidentiality/anonymity/data management, and securing respondents' informed consent for participation.

The post-traumatic stress disorder was measured using the Impact of Event Scale-Revised (IES-R-22). The IES-R was a self-administered questionnaire for determining the extent of post-traumatic stress disorder after exposure to a public health crisis within one week of exposure. The Impact event scale can be used for repeated measurements over time to monitor progress. It is an appropriate instrument to measure the subjective response to a specific traumatic event in an adult population [25]. The IES-R is available in a variety of languages and the scale is used to measure PTSD symptoms in many cultures globally [26].

A total IES-R cutoff score of 24 was used to classify PTSD as a clinical concern. The total IES-R-22 score was divided into 0–23 (normal), 24–32 (mild PTSD) 33–36 (moderate PTSD), and >37 (severe PTSD) with an internal consistency of (alpha = 0.96) [27].

We conducted a reliability analysis for the IES-R-22questionnaire (Amharic version) and found that it had a high score (Cronbach's $\alpha$ = 0.92). A socio-demographic questionnaire was used to assess the patients' background information. Clinical, psychosocial, and material factors were used to assess by yes/no answers of respondents. Social support was measured by the Oslo 3-item Social Support Scale. The social support scores range from 3–14, in that scores from 3–8 as poor, 9–11 as moderate, and 12–14 as strong social support, but for this research purpose, it was categorized as poor for scores less than nine. The scores from 9–14 were considered moderate to strong support and merged as having social support [28].

The Oslo Social Support Scale has been used in population-level studies in Ethiopia [29, 30] and showed good predictive value [31].

**Perceived stigma.** To examine the perceived stigma, respondents were asked, 'Did you feel stigmatized by the public because of your profession?' (Like being refused access to public transport, being isolated from any social affairs, and even evicted from rented homes) and the responses were Yes/No.

**History of mental illness.** To examine a history of mental illness, respondents were asked: 'Have you ever been diagnosed with mental illness and treated' and responses were yes/no.

**History of medical illness.** To examine a history of medical illness, respondents were asked: 'Did you have any medical illness?' and responses were yes/no.

**Families with medical illness.** To examine families with medical illness, respondents were asked: 'Did you have a family member who had a medical illness?' and responses were yes/no.

## Data processing and analysis

All collected data was checked for completeness and consistency and entered into Epi-data version 4.4.2 and then exported to SPSS for version 24 for analysis. We computed descriptive statistics, bivariate, and multivariate logistic regression analyses. The bivariate results may have been subject to confounding. Therefore, we conducted a multivariate analysis containing

all variables associated with PTSD. Variables for the multivariate model were selected based on a combination of known risk factors for PTSD from existing literature, and variables significantly associated with PTSD in the bivariate analyses. Those variables whose p-values <0.05 with 95CI and AOR in the multivariate model were declared predictors of PTSD.

Hosmer-Lemeshow's test (p = 0.494) was used to check model fitness. Multi-co linearity was checked to see the correlation among the independent variables by using variance inflation factor and tolerance. In this case, the value of the variance inflation factor was <10 and tolerance was greater than 0.1, which indicated that there is no dependency between independent factors.

### Ethical consideration

The ethical clearance was obtained from the ethical review committee of Debre Tabor University and a permission letter was obtained from each hospital. We received written informed consent from study participants and confidentiality was maintained by omitting personal identifiers.

### Patient and public involvement

In the current study, participants were people who were working in South Gondar Zone hospitals, Amhara, Ethiopia. Participants were not involved in the study design and recruitment. The result of this study has been disseminated to the Amhara Regional Health Bureau and each study hospital.

## Results

Of all invited 423 respondents, a total of 396 individuals completed the questionnaire. The majority of the respondents were males 274 (69.2%) and in the age group 25–30 years. Most of the participants were single 222(52.9%), orthodox followers 369(87.9%), and Amhara by ethnicity 418(99.5%). Regarding their educational status and department, the majorities of them were degree holders 292(73.7%) and nurses 230 (58.1) respectively (Table 1).

### Clinical factors of the respondents

Of all respondents, about 56(14.1%) had a history of medical problems, 140(35.4%) had families with chronic illness, and 24(6.1%) had a history of mental illness (Table 2).

### Psychosocial and material factors

About 95 (24%) of the respondents felt stigmatized because of their profession. The majority of the participants had poor social support, 233 (58.8%), and 163 (41.2%) of the respondents had moderate and strong social support. Most of the respondents 307 (77.5%) responded as they did not get standardized PPE in compacting the COVID-19 pandemic.

### Prevalence of post-traumatic stress disorder (PTSD)

Of the 423 invited health care workers, 396 (93.6%) of them completed the questionnaire. The prevalence of PTSD on health care providers in this study was 55.1% (95% CI: 50.3, 59.6).

### The severity of post-traumatic stress disorder

About 108(23.5%) of the participants had experienced severePTSD (Fig 1).

**Table 1.  Sociodemographic characteristics of health professionals working in South Gondar zone hospitals, Ethiopia, 2020(n = 396).**

| Characteristics | | Frequency | Percent |
|---|---|---|---|
| Age | <25 | 12 | 3 |
| | 25–30 | 243 | 61.4 |
| | 31–40 | 65 | 16.4 |
| | 41–50 | 55 | 13.9 |
| | >50 | 21 | 5.3 |
| Sex | Female | 122 | 30.8 |
| | Male | 274 | 69.2 |
| Marital status | Married | 200 | 50.5 |
| | Divorced | 43 | 10.9 |
| | Single | 132 | 33.3 |
| | *Others | 21 | 5.3 |
| Educational status | Diploma | 28 | 7.1 |
| | Degree | 292 | 73.7 |
| | Msc | 44 | 11.1 |
| | Specialist | 32 | 8.1 |
| Ethnicity | Amhara | 375 | 94.7 |
| | Tigray | 5 | 1.3 |
| | Oromo | 16 | 4 |
| Religious status | Orthodox | 332 | 83.8 |
| | Catholic | 10 | 2.5 |
| | Muslim | 33 | 8.3 |
| | Protestant | 11 | 2.8 |
| | Adventist | 10 | 2.5 |
| Profession | Nurse | 230 | 58.1 |
| | Physician | 77 | 19.4 |
| | Laboratory | 62 | 15.7 |
| | Pharmacists | 27 | 6.8 |
| Having children | Yes | 180 | 45.5 |
| | No | 216 | 54.5 |

Note that:

*other–separated and widowed.

## Factors associated with post-traumatic stress disorder

To determine the association of independent variables with post-traumatic stress disorder, bivariate and multivariate binary logistic regression analyses were carried out.

In this regard, the analysis was made between the dependent variable (PTSD) and independent variables including socio-demographic factors (age, gender, marital status, profession,

**Table 2.  Clinical factors of health care providers working in South Gondar zone hospitals, Amhara, Ethiopia, 2020.**

| Characteristics | Category | Frequency | percent |
|---|---|---|---|
| History of medical illness | yes | 56 | 14.1 |
| | No | 340 | 85.9 |
| Having a family member with chronic illness | yes | 140 | 35.4 |
| | No | 256 | 64.6 |
| History of mental illness | yes | 24 | 6.1 |
| | No | 372 | 93.9 |

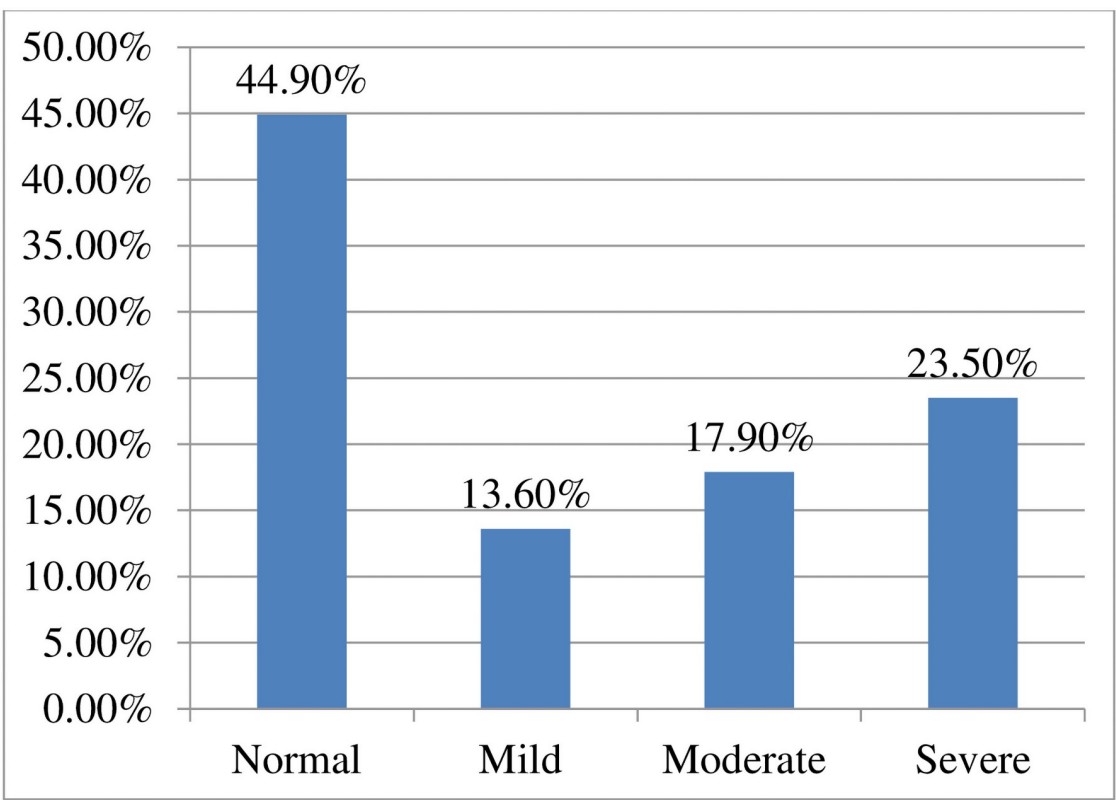

**Fig 1. Severity of post-traumatic stress disorder on health care providers in South Gondar zone hospitals, Amhara, Ethiopia, 2020.**

educational status, and having children), clinical variables (family history of medical illness, history of mental illness, having medical illness, psychosocial and material factors consist social support, perceived stigma and lack of adequate and standardized PPE supply.

On the bivariate analysis of post-traumatic stress disorder with each explanatory variable: age > 40 years, being a physician, divorced, poor social support, not getting standardized PPE, having medical problems, history of mental illness, having families with medical health problems and perceived stigma were found to be significantly associated with PTSD at a P-value <0.05 (Table 3).

All variables analyzed in bivariate analysis were taken into the multivariate analysis. In multivariate analysis, age >40 years, being a physician, lack of standardized PPE supply, having a medical illness, had perceived stigma, history of mental illness, and poor social support was significantly associated with post-traumatic stress disorder at a p-value < 0.05.

When controlling for other variables, the odds of developing post-traumatic stress disorder among health care providers were 2.57 times higher among those participants who had not standardized PPE supply as compared with those who had standardized PPE supply (AOR = 2.5 7,95CI;1.37,4.85). Those health professionals age greater than 40 years were 3.95 times more likely to develop PTSD as compared with those younger participants (AOR = 3.95, 95CI; 1.74, 8.98). The risk of developing PTSD among physicians was reduced by 15% as compared with other health professionals (AOR = 0.15, 95CI; 0.04, 0.56).

The likelihood of developing post-traumatic stress disorder among those respondents who had medical problems was 4.65 times as compared with those participants who had no medical problems (AOR = 4.65,95CI;1.65,13.12).

**Table 3. Bivariate analysis showing factors associated with PTSD on health care providers in South Gondar hospitals, Ethiopia, 2020.**

| Characteristics | Category | PTSD | | COR(95%CI) |
| --- | --- | --- | --- | --- |
| | | Yes | No | |
| Age | ≤ 30 | 124 | 131 | 1 |
| | 31–40 | 39 | 26 | 1.25(0.73,2.16) |
| | >40 | 55 | 21 | *4.02(2.20,7.36) |
| Sex | Female | 76 | 46 | 1.54(0.99,2.38) |
| | Male | 142 | 132 | 1 |
| Marital status | Married | 111 | 89 | 1 |
| | Divorced | 31 | 12 | *2.07(1.01,4.27) |
| | Single | 63 | 69 | 0.73(0.47,1.14) |
| | Others | 13 | 8 | 1.30(0.52,3.28) |
| Profession | Nurse | 136 | 94 | 0.72(0.31,1.68) |
| | Physician | 21 | 56 | *0.19(0.73,0.48) |
| | Laboratory | 43 | 19 | 1.13(0.43,2.97) |
| | Pharmacists | 18 | 9 | 1 |
| Educational status | Diploma | 19 | 9 | 1.02(0.37,2.81) |
| | Degree | 149 | 143 | 1.13(0.54,2.35) |
| | Msc | 32 | 12 | 0.88(0.36,2.19) |
| | Specialist | 18 | 14 | 1 |
| Having children | Yes | 106 | 74 | 1.33(0.89,1.98) |
| | No | 112 | 104 | 1 |
| Personal protective equipment | Yes | 39 | 50 | 1 |
| | No | 179 | 128 | *1.79(1.11,2.89) |
| Medical problems | Yes | 49 | 7 | *7.08(3.12,16.08) |
| | No | 169 | 171 | 1 |
| Families with chronic illness | Yes | 87 | 53 | *1.57(1.03,2.39) |
| | No | 131 | 125 | 1 |
| History of mental illness | Yes | 40 | 4 | *9.78(3.43,27.90) |
| | No | 178 | 174 | 1 |
| Perceived stigma | Yes | 72 | 23 | *3.32(1.97,5.59) |
| | No | 146 | 155 | 1 |
| Social support | Poor | 163 | 70 | *4.57(2.98,7.02) |
| | Moderate and Strong | 55 | 108 | 1 |

Likewise, those health care providers who felt stigmatized because of their profession were 1.97 times more likely to develop post-traumatic stress disorder as compared with those health workers who did not feel stigmatized (AOR = 1.97, 95CI;1.01, 3.85) Social support has a greater impact on the development of mental problems. Those health care providers who had poor social support were 4.41 times more likely to develop post-traumatic stress disorder as compared with those who had strong and moderate social support (AOR = 4.41,95CI;2.65,7.34).

Moreover, those participants who had a history of mental illness were 8.08 times more affected as compared with their counterparts (AOR = 8.08IC;2.18, 29.98) (Table 4).

## Discussions

Health professionals treating COVID-19 cases are at risk of developing mental health symptoms than the general population. Most importantly, the present study indicated that during the COVID-19 outbreak, healthcare workers who performed COVID-19 related tasks scored

**Table 4. Multivariable analysis showing factors associated with PTSD on health care providers in South Gondar hospitals, Ethiopia, 2020.**

| characteristics | category | PTSD | | COR(95%CI) | COR(95%CI) |
|---|---|---|---|---|---|
| | | Yes | No | | |
| Age | <30 | 124 | 131 | 1 | 1 |
| | 31–40 | 39 | 26 | 1.59(0.91,2.76) | 1.16(0.57,2.39) |
| | >40 | 55 | 21 | *2.77(1.58,4.84) | 3.95(1.74, 8.98) |
| Sex | Female | 76 | 46 | 1.54(0.99,2.38) | 1.47(0.82,2.65) |
| | Male | 142 | 132 | 1 | 1 |
| Marital status | Married | 111 | 89 | 1 | 1 |
| | Divorced | 31 | 12 | *2.07(1.01,4.27) | 0.65(0.25,1.69) |
| | Single | 63 | 69 | 0.73(0.47,1.14) | 1.08(0.52,2.23) |
| | Others | 13 | 8 | 1.30(0.52,3.28) | 0.54(0.15,1.96) |
| Profession | Nurse | 136 | 94 | 0.72(0.31,1.68) | 0.63(0.20,1.94) |
| | Physician | 21 | 56 | *0.19(0.73,0.48) | *0.15(0.04,0.56) |
| | Laboratory | 43 | 19 | 1.13(0.43,2.97) | 0.82(0.23,2.92) |
| | Pharmacists | 18 | 9 | 1 | 1 |
| Educational status | Diploma | 19 | 9 | 1.64(0.57,4.72) | 1.73(0.50,5.95) |
| | Degree | 149 | 143 | 0.81(0.39,1.69) | 1.51(0.61,3.72) |
| | Msc | 32 | 12 | 2.07(0.79,5.44) | 0.84(0.28,2.57) |
| | Specialist | 18 | 14 | 1 | 1 |
| Having children | Yes | 106 | 74 | 1.33(0.89,1.98) | 0.62(0.33,1.17) |
| | No | 112 | 104 | 1 | |
| Personal protective equipment | Yes | 39 | 50 | 1 | 1 |
| | No | 179 | 128 | *1.79(1.11,2.89) | *2.57(1.37,4.85) |
| Medical problems | Yes | 49 | 7 | *7.08(3.12,16.08) | *4.65(1.65,13.12) |
| | No | 169 | 171 | 1 | 1 |
| Families with chronic illness | Yes | 87 | 53 | *1.57(1.03,2.39) | 1.08(0.58,2.03) |
| | No | 131 | 125 | 1 | 1 |
| History of mental illness | Yes | 40 | 4 | *9.78(3.43,27.90) | *8.08(2.18,29.98) |
| | No | 178 | 174 | 1 | 1 |
| Perceived stigma | Yes | 72 | 23 | *3.32(1.97,5.59) | *1.97(1.01,3.85) |
| | No | 146 | 155 | 1 | 1 |
| Social support | Poor | 163 | 70 | *4.57(2.98,7.02) | *4.41(2.65,7.34) |
| | Moderate and Strong | 55 | 108 | 1 | 1 |

The results shown in the table are adjusted for all of the other variables listed.

NB: PTSD = post traumatic stress disorder, others = Separated and Widowed, Model fitness = (Hosmer and Lemshow Test = 0.494),

* = p = <0.05.

significantly higher on the total IES-R. This suggests that they should be the main targets of psychiatric assessment and care.

In this study, the prevalence of post-traumatic stress disorder on health care providers during the COVID-19 pandemic was found to be 55.1% (95% CI: 50.3, 59.6) and about 93 (23.5%) of care providers experienced severe post-traumatic stress disorder.

This finding was in line with the previous study conducted in China during the initial COVID-19 pandemic 53.8% [10], and the Spanish study 56.6% [32]. In contrast, the finding of this study was higher than the studies done in Korea, 40.3% [33], Israel11.5% [34], Toronto and Hamilton hospitals 13.8% [17], Singapore study 20% [35], china study 7% [36], systematic review studies 21,7% [37], 27% [38] andTaiwan5% [39]. The reasons for this may include

reduced accessibility to formal psychological support, less first–hand medical information on the outbreak, less intensive training on personal protective equipment including PPE supply shortage, and an underdeveloped health care system in Ethiopia as compared with those countries.

On the other hand, the finding of this study was lower than the previous study done on the impact of the SARS epidemic on the mental health of health care workers 82% [40]. This could be because of tool differences, socio-cultural factors. When controlling for other variables in the model, lack of a standardized PPE supply, age >40 years, having a medical illness, perceived stigma, history of mental illness, and poor social support were negatively associated with post-traumatic stress disorder. Conversely, being a physician affected PTSD positively.

Specifically, the greater likelihood of post-traumatic stress disorder occurred among those who lack a standardized PPE supply as compared with those respondents who had standardized PPE supply. This is similar to another systematic review study [41]. This might be due to that personal protective equipment is critical to protecting health care professionals' physical and mental well-being. Without this protection, they worry that they will get sick and infect others. The risk of infection, especially if it is asymptomatic, instills the fear of spreading the virus to their patients and families [42].

The likelihood of developing post-traumatic stress disorder among those respondents who had medical problems was 4.65 times as compared with those participants who had no medical problems. This is in line with the study done on evidence from a systematic review from the previous outbreaks on the potential impact of COVID-19 on mental health outcomes of health care providers [41] and a study done on mental health and psychosocial problems between medical and non-medical health workers [43]. This might be because health care providers with existing medical illnesses knew as their immunity is compromised, which made them more vulnerable and risky for the bad outcome of COVID-19 pandemic leading to psychologically more affected as compared with those individuals without medical illness.

Similarly, those health care providers who felt stigmatized because of their profession were 1.97times more likely to develop post-traumatic stress disorder as compared with those health workers who did not feel stigmatized. This was supported by previous studies [39, 40]. The evidence clearly shows that stigma could hinder HCWs of different roles and responsibilities from responding correctly. They are facing an unprecedented emergency and insidious invisible danger which, increasing workload, physical and mental stress [44]. Moreover, HCWs who expected to experience higher levels of stigma reported increased psychological distress [45]. This indicates the need to increase the community awareness about COVID-19 and the appropriate prevention strategies which in turn reduce the stigmatization of the frontline health care workers and enhances the care of patients especially COVID-19 cases.

Those health care workers age >40 years had 3.95 times the risk of developing PTSD as compared with those health care workers < = 30 years old. This was contrary to the previous studies [46–48]. This could be population, and sample size difference, scarcity of researches on older age and old ages expresses PTSD symptoms physically but less experience of emotional symptoms which decrease the detection rate. Moreover, in the current study old ages are more vulnerable to PTSD because of their risk of contracting and developing the fatal illness during the pandemic as compared with the younger adults.

Those health care providers who had poor social support were 3.89 times more likely to develop post-traumatic stress disorder as compared with those who had moderate and strong social support. This was also affirmed by previous studies [35, 40, 49]. Social support is an important factor for reducing both physical and psychological distress when faced with stressful events [50]. Moreover, respondents who had a history of mental illness were more likely affected by PTSD as compared with their counterparts. The reason could be participants with

a history of mental illness might have more neuronal damage compared with those who had no history of mental illness. As a result, they might be prone to develop PTSD during this COVID-19 pandemic. This finding was supported by the results of previous studies [51–53]. Conversely physicians were less affected by PTSD as compared with nurses. This was in agreement with the previous study [54]. This could be the fact that physicians had shorter contact and less exposure to COVID-19 patients compared with the nursing staff. This calls improving the mental wellbeing of health workers with attention to the reduction of stigma, ensuring an adequate support system such as personal protective equipment, and family support for those with a history of mental health problems.

## Limitations

The cross-sectional nature of the study design might not show temporal relationships between PTSD and its predictors. Moreover, important factors such as coping style, emotional status, and taking training about COVID-19 have been missed which could predict PTSD. Even if IES-R showed good internal consistency (Cronbach alpha = 0.92) in this study, it lacks validity in this population that it may necessitate being validated by other interested researchers.

## Conclusions

The prevalence of post-traumatic stress disorder among health care providers in this study was high. More than 1/4th of participants experienced severe post-traumatic stress disorder. Lack of standardized PPE supply, age > 40 years, having a medical illness, history of mental illness, perceived stigma, and poor social support were significantly associated with post-traumatic stress disorder at a p-value less than 0.05. Conversely, physicians were less likely to develop PTSD in this study. Therefore, regular screening for the mental health status of health care providers by trained health professionals is essential. It is also necessary to provide adequate and standardized PPE supply, giving especial emphasis to those health care providers age> 40 years, those with medical illness, history of mental illness, having poor social support, creating awareness in the community to avoid the stigma faced by health care providers who treat COVID patients. Furthermore, it would be better if other interested researchers do another study on this area by validating IES-R and including important factors that have been missed in this research.

## Supporting information

**S1 File. Questionnaire, to assess the prevalence of post-traumatic stress disorder on health professionals in the era of COVID-19 pandemic.**
(DOCX)

## Acknowledgments

The authors acknowledge Debretabor University for reviewing and approval of ethical issues. We extend our gratitude to data collectors, supervisors, and study participants for their time and effort.

## Author Contributions

**Formal analysis:** Sintayehu Asnakew, Amsalu Belete, Wubet Alebachew Bayih.

**Supervision:** Sintayehu Asnakew, Getasew Legas, Tewachew Muche Liyeh, Amsalu Belete, Getachew Yideg Yitbarek, Ermias Sisay Chanie.

**Validation:** Kalkidan Haile, Binyam Minuye Birhane, Haile Amha, Shegaye Shumet.

**Writing – review & editing:** Dejen Getaneh Feleke, Binyam Minuye Birhane.

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
