## [Decision Letter · Decision Letter 0]

11 Feb 2021

PONE-D-20-34499

Prevalence of post-traumatic stress disorder on health professionals in the era of COVID-19 pandemic, Northwest Ethiopia, 2020: multi-centred cross-sectional study.

PLOS ONE

Dear Dr. Alemayehu,

Thank you for submitting your manuscript to PLOS ONE. After careful consideration, we feel that it has merit but does not fully meet PLOS ONE’s publication criteria as it currently stands. Therefore, we invite you to submit a revised version of the manuscript that addresses the points raised during the review process.

Two experts in the field handled your manuscript, and we are appreciative of their time and contributions. Although interest was found in your study, several major concerns arose that require your attention. Please address ALL of the reviewers' comments in your revised manuscript and detail your revisions in a response-to-reviewers document. 

We look forward to receiving your revised manuscript.

Kind regards,

Frank T. Spradley

Academic Editor

PLOS ONE

2. Please include in your Methods section (or in Supplementary Information files) the participating hospitals/institutions.

Furthermore, please include additional information regarding the survey or questionnaire used in the study and ensure that you have provided sufficient details that others could replicate the analyses. For instance, if you developed a questionnaire as part of this study and it is not under a copyright more restrictive than CC-BY, please include a copy, in both the original language and English, as Supporting Information.

3.Thank you for stating the following financial disclosure:

 "he funders had no role in study design, data collection and analysis, decision to publish, or preparation of the manuscript."

5.Thank you for submitting the above manuscript to PLOS ONE. During our internal evaluation of the manuscript, we found significant text overlap between your submission and the following previously published works, some of which you are an author.

https://www.healthcarefinancenews.com/news/healthcare-workers-treating-covid-19-show-more-negative-mental-health-effects

https://www.researchsquare.com/article/rs-23828/v1

https://pubmed.ncbi.nlm.nih.gov/30343247/

https://www.newkerala.com/news/2020/60792.htm

https://www.statnews.com/2020/04/03/the-covid-19-crisis-too-few-are-talking-about-health-care-workers-mental-health/

https://www.mdpi.com/2071-1050/12/9/3834/html

http://www.ephysician.ir/2017/5212.pdf

https://link.springer.com/article/10.1186/s12992-020-00621-z?code=c5fb9c70-dc49-4d3d-8d9c-ae8d56f4a3d9&error=cookies_not_supported

Please revise the manuscript to rephrase the duplicated text, cite your sources, and provide details as to how the current manuscript advances on previous work. Please note that further consideration is dependent on the submission of a manuscript that addresses these concerns about the overlap in text with published work.

Reviewers' comments:

Reviewer's Responses to Questions

**Comments to the Author**

1. Is the manuscript technically sound, and do the data support the conclusions?

Reviewer #1: Yes

Reviewer #2: Partly

2. Has the statistical analysis been performed appropriately and rigorously? 

Reviewer #1: Yes

Reviewer #2: No

3. Have the authors made all data underlying the findings in their manuscript fully available?

Reviewer #1: No

Reviewer #2: No

4. Is the manuscript presented in an intelligible fashion and written in standard English?

Reviewer #1: No

Reviewer #2: No

5. Review Comments to the Author

Reviewer #1: This manuscript intended to reveal the prevalence and predictors of PTSD in health professionals, which addressed an important question. However, the current version suffered from several problems.

1. In introduction, predictors of PTSD under COVID-19 should be addressed with latest literature.

2. More information was expected about the participants: e.g., educational level, department.

3. 423 was calculated as sufficient sample size. However, only 396 participants were included finally, which was insufficient. How is the process of recruitment of participants? Why not more people were invited initially?

4. The instrument Oslo needs more information, such as validity, sample question, et al.

5. In regression, I wonder why not all variables were included in the analysis such as age, gender, and profession? Maybe these variables predicted PTSD significantly.

6. Very limited predictors were analyzed in the current survey, in which other potentially important factors were not discussed, such as coping style, emotional status. Authors should list this in limitations at least.

7. Overall, authors should pay more attention on the writing, including space, capitalization, punctuation, et al.

Reviewer #2: Thank you for the opportunity to review this paper. It highlights the prevalence of PTSD among health care workers responding to the COVID-19 pandemic in Amhara, Ethiopia, and also points out risk factors (e.g., access to PPE, social support, medical problems) associated with PTSD. This is a useful addition to the growing literature on COVID-19. In order to get the message across effectively, I recommend heavily proofreading with an eye for English sentence structure and grammar. In addition, I have some comments about the analysis:

Major comments on analysis:

1. What variables were in the multivariate logistic regression? Were demographic characteristics included? Things that affect the exposure and also affect PTSD should be included. I think it would be important to control for age, sex, marital status, profession, and perhaps some others, if possible. Whether to include all of the risk factors of interest in a single model should also be carefully considered. For example, perhaps social support should not be included in the assessment of marital status, as it may be a mediator of the effect of marital status on PTSD. Regardless, please indicate details about the variables in the model in both the text and a footnote to Table 3.

2. “Social support = strong” only has 11 people with PTSD. This makes it a poor reference group and perhaps its use leads to the very large confidence intervals for the social support estimates. I recommend combining the “moderate” and “strong” categories, and making this the reference group.

3. Please report either in the Methods section, or as the first comment in the Results section (not later) how many people were invited to participate and how many actually participated. In addition, please state whether people who chose not to participate were different than people who did participate in terms of key demographic characteristics.

Minor comments:

4. The abstract should clearly state that the aims of the paper were to assess the prevalence of PTSD in this population and to assess the association of certain risk factors with PTSD in this population.

5. I would be interested to see more discussion of the “perceived stigma” findings. What types of stigma have health care workers in the Amhara region faced as a result of COVID-19 (or generally)?

6. Is the IES-R validated for use in this population (e.g., Northwestern Ethiopia or Ethiopia in general)?

7. A “limitations” section should be added.

8. Be consistent with labeling the pandemic as “COVID-19” versus “Corona” or other labels.

9. The underlying data do not appear to be in the manuscript, as stated.

6. PLOS authors have the option to publish the peer review history of their article (what does this mean?). If published, this will include your full peer review and any attached files.

Reviewer #1: **Yes: **Qin Dai

Reviewer #2: No

---

## [Author Response · Author response to Decision Letter 0]

21 Apr 2021

Reviewers’ comments and authors response 

After going through the entire manuscript, you forwarded your constructive comments which have been missed to touch. Therefore, we are glad enough to express our sincerest thanks for your constructive editorial comments that could help improve novelty of our work. All the comments that have been included are highlighted in the track changes 

Editors’ comment and suggestions(1)

 Authors Response

Thank you very much, we got these comments with paramount significant and we tried to rewrite the manuscript using PLOS ONE's style requirements per the academic editors’ suggestion

 Editors’ comment and suggestions(2)

2. Please include in your Methods section (or in Supplementary Information files) the participating hospitals/institutions. Furthermore, please include additional information regarding the survey or questionnaire used in the study and ensure that you have provided sufficient details that others could replicate the analyses. For instance, if you developed a questionnaire as part of this study and it is not under a copyright more restrictive than CC-BY, please include a copy, in both the original language and English, as Supporting Information./963.

Authors Response

Yes indeed! We authors included the participating hospitals /institutions in the method section and the questionnaire has been supplied as supporting file both in Amharic and English versions 

Editors’ comment and suggestions(3)

"he funders had no role in study design, data collection and analysis, decision to publish, or preparation of the manuscript."

a. Please clarify the sources of funding (financial or material support) for your study. List the grants or organizations that supported your study, including funding received from your institution.

d. If you did not receive any funding for this study, please state: “The authors received no specific funding for this work.”

 Authors Response

Since the authors did not receive any fund for this work, we included in the previous submitted form but to make it in PLOS ONE's writing format, we amended and included in the cover letter as “The authors received no specific funding for this work”.

Editors’ comment and suggestions (4)

Authors Response

Yes indeed! The ethical statement was written in both declaration and methods section so that we deleted the part in the declaration section and only included in the method section per the editors’ comment and suggestion. 

Editors’ comment and suggestions (5)

5. Thank you for submitting the above manuscript to PLOS ONE. During our internal evaluation of the manuscript, we found significant text overlap between your submission and the following previously published works, some of which you are an author.

Authors Response

Great thanks! The whole manuscript was edited by language expert and the changes are included throughout the revised version of the manuscript and the whole manuscript has been rephrased to remove the duplicated text.

Review Comments to the Author

 Reviewer #1:

This manuscript intended to reveal the prevalence and predictors of PTSD in health professionals, which addressed an important question. However, the current version suffered from several problems.

Reviewer s’ comment and suggestions (1)

1. In introduction, predictors of PTSD under COVID-19 should be addressed with latest literature.

 Authors Response

Yes indeed! This is the part that has been missed. Thus, we have intensively searched the previous published literatures and those factors which predicted PTSD has been included in the revised manuscript. 

Reviewer s’ comment and suggestions (2)

2. More information was expected about the participants: e.g., educational level, department.

Authors Response

Yes, we authors missed the details especially about the educational status of the participants and other socidempgrpahic characteristics and now their detail is included in the revised manuscript. 

Reviewer s’ comment and suggestions (3)

3. 423 was calculated as sufficient sample size. However, only 396 participants were included finally, which was insufficient. How is the process of recruitment of participants? Why not more people were invited initially?

Authors Response

In this study there are 8 participating hospitals with a total of 736 health professionals 

We proportionally allocated the sample size to each hospital and we invited 423 participants by using simple random sampling technique. Of these, twelve (12) of the eligible participants refused to participate and five (5) of the questionnaire were discarded because of incomplete data. Finally, 396 questionnaires were included for the analysis. That is why we included the 10% percent non-response rate to the initially calculated sample size. 

Reviewer s’ comment and suggestions (4)

4. The instrument Oslo needs more information, such as validity, sample question, et al.

Authors Response

Social support was measured by the Oslo 3-item Social Support Scale. The social support scores ranges from 3-14 , in that scores from 3-8 as poor,9-11 as moderate and 12-14 as strong social support but for this research purpose it was categorized into poor for scores less than nine . The scores from 9–14 were considered moderate to strong support and merged together as social support. The OSSS-3 is the three items questionnaire assessing the possibility of social support. The three questions assessing social support in OSSS-3 are:

1. How many people are you so close to that you can count on them if you have great personal problems?

2. How much interest and concern do people show in what you do?

3. How easy is it to get practical help from neighbors if you should need it?

The Oslo Social Support Scale has been used in population level studies in Ethiopia and showed good predictive value. 

Reviewer’s comment and suggestions (5)

5. In regression, I wonder why not all variables were included in the analysis such as age, gender, and profession? Maybe these variables predicted PTSD significantly.

Authors Response

Yes indeed! all variables including socio-demographic factors (age, gender, profession, marital status, educational status, and having children), clinical variables (family history of medical illness, history of mental illness, having medical illness, psychosocial and material factors consists social support, perceived stigma and lack of adequate and standardized PPE supply were included in the bivariate analysis. However, only divorced, poor social support, not getting standardized PPE, having medical problems, history of mental illness, having families with medical health problems and perceived stigma were found to be significant with PTSD in the bivariate analysis. These variables were interred into multivariate logistic regression for further analysis to control confounding factors. Finally, perceived stigma, lack of standardized and adequate PPE supply, poor social support, having history of medical and mental illness were significantly associated with PTSD. 

Reviewer s’ comment and suggestions (6)

6. Very limited predictors were analyzed in the current survey, in which other potentially important factors were not discussed, such as coping style, emotional status. Authors should list this in limitations at least.

Authors Response

Great look! We got this comment with paramount significance i.e. not only coping style, emotional status but also training about COVID-19 has been missed that should be included as factors. So we included in the limitation part which can for possible indication of other interested research in this area 

Reviewer s’ comment and suggestions (7)

7. Overall, authors should pay more attention on the writing, including space, capitalization, punctuation, et al.

Authors Response

We have not any doubt with this comment. Hence from repeated proof-reading of the whole manuscript, we found several grammatical errors, interlinings, punctuation errors, wording and spelling errors. Thus, the whole manuscript was edited by language expert and the changes are included throughout the revised version of the manuscript.

Review Comments to the Author

Reviewer #2: Thank you for the opportunity to review this paper. It highlights the prevalence of PTSD among health care workers responding to the COVID-19 pandemic in Amhara, Ethiopia, and also points out risk factors (e.g., access to PPE, social support, medical problems) associated with PTSD. This is a useful addition to the growing literature on COVID-19. In order to get the message across effectively, I recommend heavily proofreading with an eye for English sentence structure and grammar. In addition, I have some comments about the analysis:

Reviewer‘s comment and suggestions (1)

Major comments on analysis:

1. What variables were in the multivariate logistic regression? Were demographic characteristics included? Things that affect the exposure and also affect PTSD should be included. I think it would be important to control for age, sex, marital status, profession, and perhaps some others, if possible. Whether to include all of the risk factors of interest in a single model should also be carefully considered. For example, perhaps social support should not be included in the assessment of marital status, as it may be a mediator of the effect of marital status on PTSD. Regardless, please indicate details about the variables in the model in both the text and a footnote to Table 3.

Authors Response

Yes indeed! all variables including socio-demographic factors (age, gender, profession, marital status, educational status, and having children), clinical variables (family history of medical illness, history of mental illness, having medical illness, psychosocial and material factors consists social support, perceived stigma and lack of adequate and standardized PPE supply were included in the bivariate analysis. However, only divorced, poor social support, not getting standardized PPE, having medical problems, history of mental illness, having families with medical health problems and perceived stigma were found to be significant with PTSD in the bivariate analysis. Furthermore those variables significantly associated with PTSD in the bivariate analysis were entered into the multivariate logistic regression to control the confounding variables. All these has been included in the text and table form in the revised manuscript 

Reviewer‘s comment and suggestions (2)

2. “Social support = strong” only has 11 people with PTSD. This makes it a poor reference group and perhaps its use leads to the very large confidence intervals for the social support estimates. I recommend combining the “moderate” and “strong” categories, and making this the reference group 

Authors Response

Great thanks! We authors took very important lesson and we made the corrections per the reviewer’s recommendation that is we combined moderate and strong social support in to one and made it the reference group.

 Reviewer‘s comment and suggestions (3)

3. Please report either in the Methods section, or as the first comment in the Results section (not later) how many people were invited to participate and how many actually participated. In addition, please state whether people who chose not to participate were different than people who did participate in terms of key demographic characteristics.

Authors Response

In this study there were 8 participating hospitals with a total of 736 health professionals. 

We invited 423 participants by using simple random sampling technique by proportionally allocating the sample size to each hospital. Of these, twelve (12) of the eligible participants refused to participate and five (5) of the questionnaire were discarded because of incomplete data. Finally, 396 questionnaires were included for the analysis. Those individuals who were on annual leave and severally ill were excluded. This has been included in the methods and the result section 

 Minor comments:

 Reviewer‘s comment and suggestions (4)

4. The abstract should clearly state that the aims of the paper were to assess the prevalence of PTSD in this population and to assess the association of certain risk factors with PTSD in this population.

 Authors Response

 Corrected per the reviewer suggestion 

Reviewer‘s comment and suggestions (5)

5. I would be interested to see more discussion of the “perceived stigma” findings. What types of stigma have health care workers in the Amhara region faced as a result of COVID-19 (or generally)?

 Authors Response

Sure! Stigma during the era of COVID-19 pandemic is too high as reported by the general public and this condition is even worse in case of health care providers. This is because the communities’ perception that the health care providers have close contact with the COVID-19 patents so that being far away from them by any means is recommended. In this study the participants (health professionals) were asked as “Do you perceived as you are stigmatized because you are health professional in relation to COVID-19? (Like refused access to public transport, being isolated from any social affairs, and even evicted from rented homes) and the response was yes/no. Thus significant numbers of health professionals 95(24%) were stigmatized by the public. This indicates the need to increase the communities awareness about COVID-19 and the appropriate prevention strategies instead of stigmatizing the frontline health care workers which in turn enhances the care of patients especially COVID-19 cases.

 Reviewer‘s comment and suggestions (6)

6. Is the IES-R validated for use in this population (e.g., Northwestern Ethiopia or Ethiopia in general)?

 Authors Response

Great thanks for your constructive comment you provided us. Actually IES-R is not validated in Ethiopia case in these populations as to our searching knowledge. In this study, we conducted a reliability analysis for the IES-R-22questionnaire (Amharic version) and that it had a high score (Cronbach’s α=0.92). Thus, we assumed as a good measure PTSD in this study. In fact, it would be acceptable if it was validated but we cannot do this because of time constraints and we put as a limitation to show the necessity of its validation in this population by other interested researchers. 

Reviewer‘s comment and suggestions (7)

7. A“limitations”section should be added.

 Authors Response

We added this section per the reviewer’s recommendation and suggestion as Limitation

Reviewer‘s comment and suggestions (8)

8. Be consistent with labeling the pandemic as “COVID-19” versus “Corona” or other labels.

 Authors Response

With no doubt, we took the comment and we made correction as COVID-19 pandemic through the revised manuscript 

 Reviewer‘s comment and suggestions (9)

9. The underlying data do not appear to be in the manuscript, as stated.

 Authors Response

Sure! All the data can be accessible with reasonable request from the corresponding author

---

## [Decision Letter · Decision Letter 1]

12 May 2021

PONE-D-20-34499R1

Prevalence of post-traumatic stress disorder on health professionals in the era of COVID-19 pandemic, Northwest Ethiopia, 2020: multi-centred cross-sectional study

PLOS ONE

Dear Dr. Alemayehu,

Thank you for submitting your manuscript to PLOS ONE. After careful consideration, we feel that it has merit but does not fully meet PLOS ONE’s publication criteria as it currently stands. Therefore, we invite you to submit a revised version of the manuscript that addresses the points raised during the review process.

The reviewers have remaining comments that must be addressed. In addition, the authors need to contact a professional copyeditor to proof the manuscript before resubmission.

We look forward to receiving your revised manuscript.

Kind regards,

Frank T. Spradley

Academic Editor

PLOS ONE

Reviewers' comments:

Reviewer's Responses to Questions

**Comments to the Author**

1. If the authors have adequately addressed your comments raised in a previous round of review and you feel that this manuscript is now acceptable for publication, you may indicate that here to bypass the “Comments to the Author” section, enter your conflict of interest statement in the “Confidential to Editor” section, and submit your "Accept" recommendation.

Reviewer #1: All comments have been addressed

Reviewer #2: (No Response)

2. Is the manuscript technically sound, and do the data support the conclusions?

Reviewer #1: Yes

Reviewer #2: Yes

3. Has the statistical analysis been performed appropriately and rigorously? 

Reviewer #1: Yes

Reviewer #2: Yes

4. Have the authors made all data underlying the findings in their manuscript fully available?

Reviewer #1: No

Reviewer #2: No

5. Is the manuscript presented in an intelligible fashion and written in standard English?

Reviewer #1: No

Reviewer #2: No

6. Review Comments to the Author

Reviewer #1: The manuscript intended to reveal the prevalence of PTSD in health care professionals in Ethiopia during COVID-19 pandemic. The topic is interesting and deserves to explore. However, the writing of this version suffered from several major shortcomings.

1. The English need to be improved by native speaker.

2. The relationship between IES and PTSD need more exploration in introduction.

3. The significance of this investigation to the current pandemic need to be clearly addressed in the introduction.

4. How to health care professionals? This point need to be addressed in introduction or methods.

5. In introduction, latest literature referring to the PTSD in health care professionals especially during the current pandemic should be systematically reviewed.

6. 423 participants are insufficient for the sample size, which indicated that the current sample size is in a low power of effect size. Any statistic methods carried out to improve the effect size? Or any other methods could be carried out to solve the problem?

7. The format of Figure and Table need to be attended, especially the capitalization and the accuracy of English (severe instead of sever)

8. Model information about the regression should be given in the results.

Reviewer #2: Thank you for the opportunity to review the revised version of this manuscript. It is greatly improved. After the authors clarified the multivariate regression methods, I have do have some remaining concerns about that set of analyses.

Major comments:

1. I am interested in seeing the bivariate results in addition to the multivariate results. These should all be displayed. I suggest one table with bivariate results, and another table with results of the multivariate analysis. Even the “non-significant” results should be shown, along with 95% CI. This would allow the reader to assess the size and precision of each association.

2. In the multivariate analysis, I think that it is more important to choose variables to add in the multivariable model based on conceptual reasons/based on existing literature, rather than p<0.05 in the bivariate analyses. P>0.05 might be driven by small sample size or other statistical reasons. I think it would be particularly important to include age and gender in the multivariable model, since they are strongly associated with PTSD and many of the other factors in the model.

3. When interpreting results of the multivariable model, add “when controlling for the other variables in the model.” For example, “The odds of developing post-traumatic stress disorder among health care providers were 2.27 times higher among those participants who had not standardized PPE supply as compared with those who had standardized PPE supply, *controlling for the other variables in the model* (AOR=2.27,95CI;1.29,3.98).” Similarly, in the discussion, it should be made clear that the results described are from the multivariate (adjusted) models.

4. The following phrase on page 6 should be revise for clarity: “Factors associated with post-traumatic stress disorder were selected during the bivariate analysis with p<0.05 for further analysis in the multivariable logistic regression analysis. In the multivariable logistic regression analysis, variables with p<0.05 at 95% CI with adjusted OR were considered statistically significant.”

5. I suspect that marital status is not significantly associated with PTSD in the multivariate model because social support is a mediator of this relationship (i.e., being unmarried leads to poor social support, which then leads to PTSD), and social support is controlled for. This does not have to be mentioned in the discussion, but could be.

Minor comments:

1. On page 3, several previous studies of the prevalence of PTSD are mentioned. Were these all among health care workers? Nurses, specifically? Please specify.

2. Is South Gondar 666 *kilometers* from Addis Ababa? Miles? Please specify.

3. In describing the sample size calculations, does P refer to prevalence? Also, I think that specifying Z=1.96, 95% CI, and alpha=0.05 is repetitive. I think just specifying a standard normal distribution and alpha=0.05 would be enough.

4. The sentence about the number of participants from each hospital could be rewritten as, “Health professionals were from Debretabor (N=325), Andabet (N=62), Estie (N=55)..." for clarity.

5. Please specify that 396 participants, not 396 questionnaires, were included.

6. It is noted that, “All health professionals working in South Gondar Zone hospitals were included, and those participants who were on annual leave and severely ill were excluded.” This is confusing following the statement that there were 396 participants included. Perhaps this comment should be moved to the beginning of the paragraph, if it is referring to initial eligibility criteria?

7. When describing the tools used to measure PTSD, social support, etc. the authors seems to discuss each tool, and then describe them all again in the same order. Can this section be restructured to have a single section on each tool?

8. In “data processing and analysis” please change “descriptive, bivariate, and multivariate logistic regression” to “descriptive statistics, and bivariate and multivariate logistic regression” (i.e., the word “statistics” is missing.)

9. Page 9: State that the *prevalence*, not magnitude, of PTSD was 55.1%. Magnitude can be confused with severity.

10. Unless specified otherwise by editors, limitations may be better off before the conclusions.

11. I suggest removing the detail about the number of participants invited from the abstract (and only leave the number that actually were included), as it is distracting from the primary message of the abstract. There is sufficient detail in the main manuscript.

7. PLOS authors have the option to publish the peer review history of their article (what does this mean?). If published, this will include your full peer review and any attached files.

Reviewer #1: No

Reviewer #2: No

---

## [Author Response · Author response to Decision Letter 1]

1 Jun 2021

Reviewers’ comments and authors response 

In the first place we would like to forward our great thanks for your constructive comments and suggestions for the improvement of the manuscript. Thus, all the comments that have been included are highlighted in the track changes. 

Reviewer #1: The manuscript intended to reveal the prevalence of PTSD in health care professionals in Ethiopia during COVID-19 pandemic. The topic is interesting and deserves to explore. However, the writing of this version suffered from several major shortcomings.

 Reviewer 1 comments and suggestions (1)

1. The English need to be improved by native speaker.

 Response 

Great thanks! We invited the language expert and the manuscript has been edited and the changes are included throughout the revised version of the manuscript. 

 Reviewer 1 comments and suggestions (2)

2. The relationship between IES and PTSD need more exploration in introduction.

Response 

The relationships between IES and PTSD have been stated per the reviewer’s suggestion and comment 

 Reviewer 1 comments and suggestions (3)

3. The significance of this investigation to the current pandemic need to be clearly addressed in the introduction.

 Response 

Yes indeed! This is the part that we missed and correction has been made per the reviewer’s comments and suggestions 

 Reviewer 1 comments and suggestions (4)

4. How to health care professionals? This point need to be addressed in introduction or methods.

 Response 

We included this point in the introduction part per the reviewer’s suggestions and comments 

 Reviewer 1 comments and suggestions (5)

5. In introduction, latest literature referring to the PTSD in health care professionals especially during the current pandemic should be systematically reviewed.

 Response

Certainly! We reviewed the newly published articles and included them in the introduction part.

 Reviewer 1 comments and suggestions (6)

6. 423 participants are insufficient for the sample size, which indicated that the current sample size is in a low power of effect size. Any statistic methods carried out to improve the effect size? Or any other methods could be carried out to solve the problem?

 Response 

We want to say sorry for the confusion. In fact the total health professionals in the eight hospitals were 736. We then calculated the sample using single proportion formula giving the final sample size of 423and the response rate was 94.8%. Thus, the sample size, 423 is assumed to be sufficient for the study populations of 736 which could also be a representative 

Reviewer 1 comments and suggestions (7)

7. The format of Figure and Table need to be attended, especially the capitalization and the accuracy of English (severe instead of sever) 

 Response

Yes indeed! We got the mistake and corrected in the revised figure 

 Reviewer 1 comments and suggestions (8)

8. Model information about the regression should be given in the results.

 Response

Great thanks! This comment has been also raised by reviewer2 so that we revised the regression and put in the revised manuscript 

Reviewer #2: Thank you for the opportunity to review the revised version of this manuscript. It is greatly improved. After the authors clarified the multivariate regression methods, I have do have some remaining concerns about that set of analyses.

Major comments:

 Reviewer 2 comments and suggestions (1)

1. I am interested in seeing the bivariate results in addition to the multivariate results. These should all be displayed. I suggest one table with bivariate results, and another table with results of the multivariate analysis. Even the “non-significant” results should be shown, along with 95% CI. This would allow the reader to assess the size and precision of each association.

 Response 

Certainly! We revised the analysis and we put the analysis as bivariate and multivariate in separate tables in the revised manuscript 

 Reviewer 2 comments and suggestions (2)

2. In the multivariate analysis, I think that it is more important to choose variables to add in the multivariable model based on conceptual reasons/based on existing literature, rather than p<0.05 in the bivariate analyses. P>0.05 might be driven by small sample size or other statistical reasons. I think it would be particularly important to include age and gender in the multivariable model, since they are strongly associated with PTSD and many of the other factors in the model.

 Response

Sure! We took the comments and suggestions given by the reviewer and we took all variables analyzed in bivariate into multivariate analysis. In this case, age was re categorized because in the cell there was 1.3 % frequency which was < 5%. Thus, age was significantly associated with PTSD.

Reviewer 2 comments and suggestions (3)

3. When interpreting results of the multivariable model, add “when controlling for the other variables in the model.” For example, “The odds of developing post-traumatic stress disorder among health care providers were 2.27 times higher among those participants who had not standardized PPE supply as compared with those who had standardized PPE supply, *controlling for the other variables in the model* (AOR=2.27,95CI;1.29,3.98).” Similarly, in the discussion, it should be made clear that the results described are from the multivariate (adjusted) models.

 Response

Corrected per the reviewer suggestions and comments 

Reviewer 2 comments and suggestions (4)

4. The following phrase on page 6 should be revise for clarity: “Factors associated with post-traumatic stress disorder were selected during the bivariate analysis with p<0.05 for further analysis in the multivariable logistic regression analysis. In the multivariable logistic regression analysis, variables with p<0.05 at 95% CI with adjusted OR were considered statistically significant.”

 Response

Great thanks! we found it as confusing and correction has been made per reviewers suggestion

 Reviewer 2 comments and suggestions (5)

5. I suspect that marital status is not significantly associated with PTSD in the multivariate model because social support is a mediator of this relationship (i.e., being unmarried leads to poor social support, which then leads to PTSD), and social support is controlled for. This does not have to be mentioned in the discussion, but could be.

 Response

Certainly! We took great lesson here so that we checked multi-co linearity between independent variables including marital status and social support using VIF and tolerance. However, no multicolinearity was found between independent factors. 

Minor comments:

 Reviewer 2 comments and suggestions (1)

1. On page 3, several previous studies of the prevalence of PTSD are mentioned. Were these all among health care workers? Nurses, specifically? Please specify.

 Response

Sure! The studies were among all health care workers

 Reviewer 2 comments and suggestions (2)

2. Is South Gondar 666 *kilometers* from Addis Ababa? Miles? Please specify.

 Response

Sorry for the confusion, it is kilometers and correction has been made on the revised manuscript

 Reviewer 2 comments and suggestions (3)

3. In describing the sample size calculations, does P refer to prevalence? Also, I think that specifying Z=1.96, 95% CI, and alpha=0.05 is repetitive. I think just specifying a standard normal distribution and alpha=0.05 would be enough.

 Response

Certainly! Redundancy of idea was noted and we corrected it per the reviewer’s suggestions and comments 

 Reviewer 2 comments and suggestions (4)

4. The sentence about the number of participants from each hospital could be rewritten as, “Health professionals were from Debretabor (N=325), Andabet (N=62), Estie (N=55)..." for clarity.

 Response

Corrected per the reviewers comments and suggestions 

Reviewer 2 comments and suggestions (5)

5. Please specify that 396 participants, not 396 questionnaires, were included.

 Response

Corrected as participants in the revised manuscript 

Reviewer 2 comments and suggestions (6)

6. It is noted that, “All health professionals working in South Gondar Zone hospitals were included, and those participants who were on annual leave and severely ill were excluded.” This is confusing following the statement that there were 396 participants included. Perhaps this comment should be moved to the beginning of the paragraph, if it is referring to initial eligibility criteria?

 Response

Great look! It was really confusing, thus we made the necessary correction per the reviewers suggestions 

Reviewer 2 comments and suggestions (7)

7. When describing the tools used to measure PTSD, social support, etc. the authors seems to discuss each tool, and then describe them all again in the same order. Can this section be restructured to have a single section on each tool?

 Response

Sure! It was really redundant and confusing so that correction has been made by classifying as “data source and measurement” and operational definitions” 

Reviewer 2 comments and suggestions (8)

8. In “data processing and analysis” please change “descriptive, bivariate, and multivariate logistic regression” to “descriptive statistics, and bivariate and multivariate logistic regression” (i.e., the word “statistics” is missing.)

 Response

Great thanks! Correction has been made 

Reviewer 2 comments and suggestions (9)

9. Page 9: State that the *prevalence*, not magnitude, of PTSD was 55.1%. Magnitude can be confused with severity.

 Response

We replaced magnitude with prevalence per the reviewers comment 

Reviewer 2 comments and suggestions (10)

10. Unless specified otherwise by editors, limitations may be better off before the conclusions.

 Response

We took the comment with paramount significance and we made it before the conclusion in the revised manuscript 

Reviewer 2 comments and suggestions (11)

11. I suggest removing the detail about the number of participants invited from the abstract (and only leave the number that actually were included), as it is distracting from the primary message of the abstract. There is sufficient detail in the main manuscript.

 Response

Great! Corrected per the reviewers comment and suggestions

---

## [Decision Letter · Decision Letter 2]

6 Jul 2021

PONE-D-20-34499R2

Prevalence of post-traumatic stress disorder on health professionals in the era of COVID-19 pandemic, Northwest Ethiopia, 2020: multi-centred cross-sectional study

PLOS ONE

Dear Dr. Alemayehu,

Thank you for submitting your manuscript to PLOS ONE. After careful consideration, we feel that it has merit but does not fully meet PLOS ONE’s publication criteria as it currently stands. Therefore, we invite you to submit a revised version of the manuscript that addresses the points raised during the review process.

We look forward to receiving your revised manuscript.

Kind regards,

Frank T. Spradley

Academic Editor

PLOS ONE

Journal Requirements:

Reviewers' comments:

Reviewer's Responses to Questions

**Comments to the Author**

1. If the authors have adequately addressed your comments raised in a previous round of review and you feel that this manuscript is now acceptable for publication, you may indicate that here to bypass the “Comments to the Author” section, enter your conflict of interest statement in the “Confidential to Editor” section, and submit your "Accept" recommendation.

Reviewer #1: All comments have been addressed

Reviewer #2: (No Response)

2. Is the manuscript technically sound, and do the data support the conclusions?

Reviewer #1: Yes

Reviewer #2: Yes

3. Has the statistical analysis been performed appropriately and rigorously? 

Reviewer #1: Yes

Reviewer #2: Yes

4. Have the authors made all data underlying the findings in their manuscript fully available?

Reviewer #1: No

Reviewer #2: No

5. Is the manuscript presented in an intelligible fashion and written in standard English?

Reviewer #1: No

Reviewer #2: No

6. Review Comments to the Author

Reviewer #1: Authors have done a good job in revision. I have only one suggestions: The language of this manuscript still needs improve, please improve the language substantially.

Reviewer #2: Thank you for the opportunity to review a revised version of this manuscript. It is improved from the previous version. Thanks to the authors for adding the bivariate results. Although I list several comments below, they are minor in nature. I would also like to note that copy-editing is necessary.

1. First sentence is confusing. I don’t think SARS should be mentioned, as it takes the focus away from COVID-19.

2. In the first paragraph, “the election” is referenced. Which election?

3. Information about IES-R was added to the introduction in this version. In my opinion, that should be saved for the methods section.

4. Fix the last sentence of the introduction: “magnitude and associated factors” is confusing. Maybe “prevalence of PTSD, as well as risk factors for PTSD.”

5. I want to clarify point 2 in my previous review (“In the multivariate analysis, I think that it is more important to choose variables to add in the multivariable model based on conceptual reasons/based on existing literature, rather than p<0.05. P<0.05 might be driven by small sample size or other statistical reasons”). I think p<0.05 alone that this is not a good reason to choose variables for the multivariate model. The decision should also be based on existing literature. Perhaps you can say, “the bivariate results may have been subject to confounding. Therefore, we conducted a multivariate analysis containing all variables associated with PTSD. Variables for the multivariate model were selected based on a combination of known risk factors for PTSD from existing literature, and variables significantly associated with PTSD in the bivariate analyses.” Also, remove “not too large” from this section.

6. In Table 4, there should be a footnote stating that the results shown are adjusted for all of the other variables listed. I would also suggest adding, “when controlling for other variables in the model” to paragraph 2 in the Discussion. I think this information is important for interpreting the results from the multivariate (adjusted) model.

7. Thank you for your revision in response to point 7 on my last review (“When describing the tools used to measure PTSD, social support, etc. the authors seems to discuss each tool, and then describe them all again in the same order. Can this section be restructured to have a single section on each tool?”). I appreciate that you have “data sources and measurement” and then “operational definitions.” It would still be my preference for easier reading to have each tool described once (measurement and operational definition together). As previously noted, some of this information is currently in the introduction, as well.

Wording comments:

8. Page 9: Instead of “found to be significant,” should be “found to be significantly associated with PTSD.”

9. On page 12, “magnitude” should be replaced with “prevalence.”

10. On page 12, “corona outbreak” should be replaced with “COVID-19 pandemic.”

11. On page 13, I am not sure what is meant by “conversely, being a physician affected PTSD positively.” Do you mean “being a physician made PTSD less likely”?

12. Replace “likely hood” with “likelihood” throughout.

7. PLOS authors have the option to publish the peer review history of their article (what does this mean?). If published, this will include your full peer review and any attached files.

Reviewer #1: **Yes: **Qin Dai

Reviewer #2: No

---

## [Author Response · Author response to Decision Letter 2]

10 Jul 2021

Reviewers’ comments and authors response 

In the first place we would like to forward our great thanks for your constructive comments and suggestions for the improvement of the manuscript. In addition, we would like to thank for scientific lesson we got throughout your comments and suggestions. Thus, all the comments that have been included are highlighted in the track changes. 

Reviewer #1: Authors have done a good job in revision. I have only one suggestion: The language of this manuscript still needs improve, please improve the language substantially. 

.

 Response 

Great thanks again! From repeated proof-reading, we found several grammatical errors, punctuations errors, wording and spelling errors etc. Therefore we invited an expert with Masters of Arts in English; we have tried our best to thoroughly copyedit the manuscript for appropriate English language usage. Thus the changes are found throughout the revised version of the manuscript. 

Reviewer #2: Thank you for the opportunity to review a revised version of this manuscript. It is improved from the previous version. Thanks to the authors for adding the bivariate results. Although I list several comments below, they are minor in nature. I would also like to note that copy-editing is necessary.

Reviewer 2 comments and suggestions (1)

1. First sentence is confusing. I don’t think SARS should be mentioned, as it takes the focus away from COVID-19.

 Response 

Great thanks! Correction has been made per the reviewer’s suggestions 

 Reviewer 2 comments and suggestions (2)

2. In the first paragraph, “the election” is referenced. Which election?

 Response

Great! It is the sixth national election of Ethiopia 

Reviewer 2 comments and suggestions (3) 

3. Information about IES-R was added to the introduction in this version. In my opinion, that should be saved for the methods section.

 Response

Great thanks! We removed the information about IES-R in the introduction part and added in the method section per the reviewer’s suggestions and recommendations. 

4. Fix the last sentence of the introduction: “magnitude and associated factors” is confusing. Maybe “prevalence of PTSD, as well as risk factors for PTSD.”

Reviewer 2 comments and suggestions (4) 

 Response

Yes! We replaced magnitude with Prevalence per the reviewer’s suggestions and comments 

Reviewer 2 comments and suggestions (5) 

5. I want to clarify point 2 in my previous review (“In the multivariate analysis, I think that it is more important to choose variables to add in the multivariable model based on conceptual reasons/based on existing literature, rather than p<0.05. P<0.05 might be driven by small sample size or other statistical reasons”). I think p<0.05 alone that this is not a good reason to choose variables for the multivariate model. The decision should also be based on existing literature. Perhaps you can say, “the bivariate results may have been subject to confounding. Therefore, we conducted a multivariate analysis containing all variables associated with PTSD. Variables for the multivariate model were selected based on a combination of known risk factors for PTSD from existing literature, and variables significantly associated with PTSD in the bivariate analyses.” Also, remove “not too large” from this section.

 Response

 Corrected per the suggestions and comments 

Reviewer 2 comments and suggestions (6)

6. In Table 4, there should be a footnote stating that the results shown are adjusted for all of the other variables listed. I would also suggest adding, “when controlling for other variables in the model” to paragraph 2 in the Discussion. I think this information is important for interpreting the results from the multivariate (adjusted) model.

 Response

Sure! Corrected per the reviewers suggestions and comments in the revised manuscript. 

 Reviewer 2 comments and suggestions (7) 

7. Thank you for your revision in response to point 7 on my last review (“When describing the tools used to measure PTSD, social support, etc. the authors seems to discuss each tool, and then describe them all again in the same order. Can this section be restructured to have a single section on each tool?”). I appreciate that you have “data sources and measurement” and then “operational definitions.” It would still be my preference for easier reading to have each tool described once (measurement and operational definition together). As previously noted, some of this information is currently in the introduction, as well.

 Response

Great thanks! We made the necessary correction per the comments and suggestions in the revised manuscript.

Wording comments:

 Reviewer 2 comments and suggestions (8) 

8. Page 9: Instead of “found to be significant,” should be “found to be significantly associated with PTSD.

 Response

Corrected per the recommendations of the reviewers 

Reviewer 2 comments and suggestions (9)

9. On page 12, “magnitude” should be replaced with “prevalence.”

 Response

Corrected per the reviewers recommendations

Reviewer 2 comments and suggestions (10)

10. On page 12, “corona outbreak” should be replaced with “COVID-19 pandemic.”

 Response

Corrected per the reviewers recommendations

Reviewer 2 comments and suggestions (11) 

11. On page 13, I am not sure what is meant by “conversely, being a physician affected PTSD positively.” Do 

you mean “being a physician made PTSD less likely”?

 Response

Certainly! It is really confusing and unclear sentences so that we made the corrections in the revised manuscript as “conversely physician were less affected by PTSD as compared with nurses” 

Reviewer 2 comments and suggestions (12) 

12. Replace “likely hood” with “likelihood” throughout.

Response 

Corrected as “likelihood” throughout the revised manuscript

---

## [Editor Report · Decision Letter 3]

15 Jul 2021

Prevalence of post-traumatic stress disorder on health professionals in the era of COVID-19 pandemic, Northwest Ethiopia, 2020: multi-centred cross-sectional study

PONE-D-20-34499R3

Dear Dr. Alemayehu,

We’re pleased to inform you that your manuscript has been judged scientifically suitable for publication and will be formally accepted for publication once it meets all outstanding technical requirements.

Kind regards,

Frank T. Spradley

Academic Editor

PLOS ONE

---

## [Editor Report · Acceptance letter]

6 Sep 2021

PONE-D-20-34499R3 

Prevalence of post-traumatic stress disorder on health professionals in the era of COVID-19 pandemic, Northwest Ethiopia, 2020: a multi-centered cross-sectional study 

Dear Dr. Asnakew:

I'm pleased to inform you that your manuscript has been deemed suitable for publication in PLOS ONE. Congratulations! Your manuscript is now with our production department. 

Kind regards, 

on behalf of

Dr. Frank T. Spradley 

Academic Editor

PLOS ONE